# γ-BaFe$_2$O$_4$: a fresh playground for room temperature multiferroicity

Fabio Orlandi ®[1] ✉, Davide Delmonte ®[2], Gianluca Calestani[3], Enrico Cavalli[3], Edmondo Gilioli[2], Vladimir V. Shvartsman ®[4], Patrizio Graziosi[5], Stefano Rampino ®[2], Giulia Spaggiari[2,6], Chao Liu[7,8], Wei Ren ®[7], Silvia Picozzi[8], Massimo Solzi ®[6], Michele Casappa ®[2,3] & Francesco Mezzadri ®[2,3] ✉

Multiferroics, showing the coexistence of two or more ferroic orderings at room temperature, could harness a revolution in multifunctional devices. However, most of the multiferroic compounds known to date are not magnetically and electrically ordered at ambient conditions, so the discovery of new materials is pivotal to allow the development of the field. In this work, we show that BaFe$_2$O$_4$ is a previously unrecognized room temperature multiferroic. X-ray and neutron diffraction allowed to reveal the polar crystal structure of the compound as well as its antiferromagnetic behavior, confirmed by bulk magnetometry characterizations. Piezo force microscopy and electrical measurements show the polarization to be switchable by the application of an external field, while symmetry analysis and calculations based on density functional theory reveal the improper nature of the ferroelectric component. Considering the present findings, we propose BaFe$_2$O$_4$ as a Bi- and Pb-free model for the search of new advanced multiferroic materials.

Multiferroics are materials showing the coexistence of two or more primary ferroic orderings[1], such as ferroelectricity, ferri/ferro-magnetism (antiferromagnetism), ferroelasticity, and ferrotoroidicity[2]. In particular, magneto-electric materials have attracted considerable interest of the research community due to the possible transformative advances they can bring in logic-memory devices, by reducing the energy consumption or increasing information density, taking advantage of both electric and magnetic domains[3]. From this perspective, the observation in early 2000's of the possibility to control, through the application of an electric field, antiferromagnetic domains in 600-nm-thick BiFeO$_3$ films[4] motivated the community even more.

Beside the mutual control of ferroelectricity and ferromagnetism, the coexistence of different ferroic orders in the same device opens the route to new applications, such as electric field-controlled spintronic devices, tunnel magnetoresistance sensors, and spin valves, just to name few[5]. Noteworthy, the use of antiferromagnetic ferroelectrics has been suggested as appealing in recent years, since magnetoelectric coupling can be achieved through exchange bias in a ferromagnetic/ antiferromagnetic magnetoelectric composite[6,7]. In addition, antiferromagnets are characterized by lower switching energy, allowing power saving, and intrinsic reduced volatility, suitable for the realization of next-generation magneto-electric spin field effect transistors[8] or magnetoelectric random access memories[9].

[1]ISIS Facility, Rutherford Appleton Laboratory, Harwell Campus, Didcot OX11 0QX, UK. [2]IMEM-CNR, Parco Area delle Scienze 37/A, 43124 Parma, Italy. [3]Department of Chemistry, Life Sciences and Environmental Sustainability, University of Parma, Parco Area delle Scienze 17/A, 43124 Parma, Italy. [4]Institute for Materials Science and Center for Nanointegration Duisburg-Essen (CENIDE), University of Duisburg-Essen, Universitätsstrasse 15, 45141 Essen, Germany. [5]CNR-ISMN, Via P. Gobetti 101, 40129 Bologna, Italy. [6]Department of Mathematical, Physical and Computer Sciences, University of Parma, Parco Area delle Scienze 7/A, Parma, Italy. [7]Physics Department, International Center of Quantum and Molecular Structures, Materials Genome Institute, State Key Laboratory of Advanced Special Steel, Shanghai Key Laboratory of High Temperature Superconductors, Shanghai University, Shanghai 200444, China. [8]Consiglio Nazionale delle Ricerche (CNR-SPIN), Unità di Ricerca presso Terzi c/o Università "G. D'Annunzio", 66100 Chieti, Italy. ✉e-mail: fabio.orlandi@stfc.ac.uk; francesco.mezzadri@unipr.it

To achieve such goals, there is the need of suitable magneto-electric materials with stable electric and magnetic orderings in ambient conditions, large magnetoelectric coupling, possibly compatible with the current silicon-based architectures and environmentally friendly[10]. These conditions are hard to obtain in a single-phase material[11] with most candidate to date having transition temperatures below room temperature, with only few exceptions, or having weak magneto-electric couplings, such as barium (and strontium) hexaferrites and their partially Co-, Al- or In-substituted counterparts[12–16], Aurivillius phases[17] and perovskite lead-based compounds[18,19].

Bismuth ferrite (BiFeO₃) fulfils all the requirements displaying at room temperature both a large ferroelectric polarization[20–22], ascribed to the $Bi^{3+}$ $6s^2$ electrons stereochemical effect[23], and a very long periodicity cycloidal antiferromagnetic structure[24]. For these reasons, it is currently the most, and almost the only, used material for magneto-electric device development and testing. Nevertheless, considering that each different application requires materials with specific features, the discovery of new magnetoelectric multiferroics is pivotal to allow the development of the field.

In this work, we study the room temperature ferroelectric and magnetic properties of γ-BaFe₂O₄, a stuffed tridymite-type compound, in which the $Si^{4+}$ tetravalent ions are replaced by the lower-valent $Fe^{3+}$ ions and electroneutrality is achieved by the inclusion of $Ba^{2+}$ within the cavities of the framework. Through careful and detailed characterizations, we show that γ-BaFe₂O₄ manifests improper ferroelectricity, without the presence of Pb and Bi which are health-threatening elements, and antiferromagnetism at room temperature. Both orderings possess very high characteristic temperatures ($T_{FE} > 1038$ K and $T_N = 890$ K) making them very robust at ambient conditions. Moreover, these characteristics are maintained in thin film form. Indeed, we show preliminary results indicating a facile deposition of γ-BaFe₂O₄ on silicon substrates.

Our results strongly suggest γ-BaFe₂O₄ as a new candidate for application as a multiferroic material. Furthermore, considering that many compositions are known to share the same $AB_2O_4$ structure (with A = Pb, Sr, Ca, Ba and B = Al, Fe, Ga)[25–31], the present study suggests that the stuffed tridymite class of materials, through suitable chemical substitutions at the A and B sites, can play a relevant role as a hitherto unexplored host for multiferroic properties at RT.

## Results
### Structural characterization
The single crystal structural determinations, considering the complex twinning of the single crystals (Fig. S1) due to the phase transition encountered by the material during synthesis[32], allowed observing relevant features of γ-BaFe₂O₄ previously not described. Solution and refinement in the centrosymmetric space group *Cmcm* (n. 63) did not allow obtaining a reliable model, with unrealistic sevenfold coordination of the iron ions and a high R1 reliability factor of ≈14%.

On the other hand, when the polar *Cmc2₁* space group (n. 36) was used, a clear improvement of the reliability factors and normalization of the structural features were observed, in agreement with previous reports[33] (see Table S1 in the SI for details of the structural refinement). Such approach was effectively exploited for the correct space group assignment in other ferroelectric compounds in which, due to the small deviation from the centrosymmetric model, a comparative approach taking into account the physical properties of the system was required[15,16,34]. The crystal structure of γ-BaFe₂O₄ displays the "stuffed tridymite" framework with an up-up-up-up-up-down (UUUUUD) pattern of tetrahedra around the triangular cavities running along the [100] direction (see Fig. 1a). However, differently, to previous structural determinations, our characterization identified a -0.03 Å shift of all the $Fe^{3+}$ ions from the center of the coordination polyhedra. As shown in Fig. 1b, an ordered pattern of short (and long) Fe−O bonds (in the range 1.827(12)−1.890(11) Å) pointing along the

[001] direction (i.e., along the polar axis of the *Cmc2₁* structure) is observed. Due to the severe twinning, it is not possible to conclude whether such off-centering takes place along the positive or negative direction of the c axis, however, the present structural determination clearly indicates the presence of a net dipole moment in the system and, consequently, possible ferroelectric properties.

To characterize in detail the oxygen positions and the room temperature magnetic structure of γ-BaFe₂O₄ neutron powder diffraction data were collected on the WISH diffractometer (ISIS-UK) in the 300–1038 K temperature range. The room temperature dataset indicates the presence of strong reflections violating the *C* centering of the nuclear space group [k = (1,0,0)]. These extra reflections decrease monotonously with temperature (see Fig. S2), eventually disappearing above 900 K, thus suggesting that they are magnetic in nature. The high temperature, paramagnetic data set collected at 1038 K was refined with the *Cmc2₁* nuclear model obtained from the single crystal diffraction data. Very good agreement between observed and calculated data has been achieved, confirming the structural model obtained from single crystal X-ray diffraction, within the 0.5 % average error on the atom coordinates. The refinement of the oxygen atoms' occupancies, thanks to the high scattering length of the oxygen atoms with respect to the X-ray scattering factor (quite low if related to the ones of Ba and Fe), allowed to rule out any significant deviation from the stoichiometric ratio. Excluding temperature-induced changes in the cell parameters and bond lengths, no other changes of the nuclear structure were observed in the explored temperature range. Rietveld plots, structural parameters, and reliability factors are reported in the SI in Figs. S3 and Table S2.

The magnetic structure of γ-BaFe₂O₄ has been solved within the $P_Cca2_1$ magnetic space group corresponding to the mY₄ irreducible representation and related to the parent *Cmc2₁* structure through the transformation {(0,1,0),(1,0,0),(0,0,−1)} with origin at (1/4, 1/4, 0)[35]. The magnetic structure is a G-type ordering with each $Fe^{3+}$ moment coupled antiferromagnetically with the nearest neighbor ones. The moments lie along the *b*-axis within error (*a*-axis of the parent *Cmc2₁* structure) and the refined moment size at 300 K is 3.740(4) μB $Fe^{-1}$. It is worth stressing that the magnetic space group does not allow any ferromagnetic component. Rietveld plots (Fig. S4), atomic parameters, reliability factors (Table S3) and mcif file are reported as supporting information.

### Magnetic and ferroelectric characterization
The magnetic properties of γ-BaFe₂O₄ were studied as a function of temperature and magnetic field by SQUID vibrating sample magnetometry in the 300–1000 K temperature range. Figure 2a shows the zero-field cooling warming (ZFC) and field-cooled cooling (FCC) measurement in an applied field of 2 T indicating a magnetic transition at $T_N$ ~ 890 K in agreement with the neutron diffraction data. The irreversibility between the two measurements suggests the presence of a weak ferromagnetic moment. This is in apparent contrast with the neutron diffraction experiments, showing an AFM material without any ferromagnetic canting. Detailed magnetometry measurements performed at different fields and on different sample batches (see Fig. S5) indicate that the weak ferromagnetic moment is strongly sample-dependent and can be correlated with the presence of iron-containing impurities with particular regards to barium hexaferrite BaFe₁₂O₁₉ ($T_C$ ~ 720 K)[36] and BaFe₄O₇ ($T_N = 850$ K)[37]. It should be noticed that even in the best quality samples, where no spurious phases were detected by diffraction techniques, very small amounts of impurities can hide the AFM character of the material.

Figure 2b shows the magnetization versus the applied field at 300, 800, 860, and 1000 K. The loops possess a linear behavior in the high field regime compatible with a global antiferromagnetic character. The weak hysteretic behavior at low field is consistent with the presence of Ba hexaferrite and other Fe-containing impurities below the detection

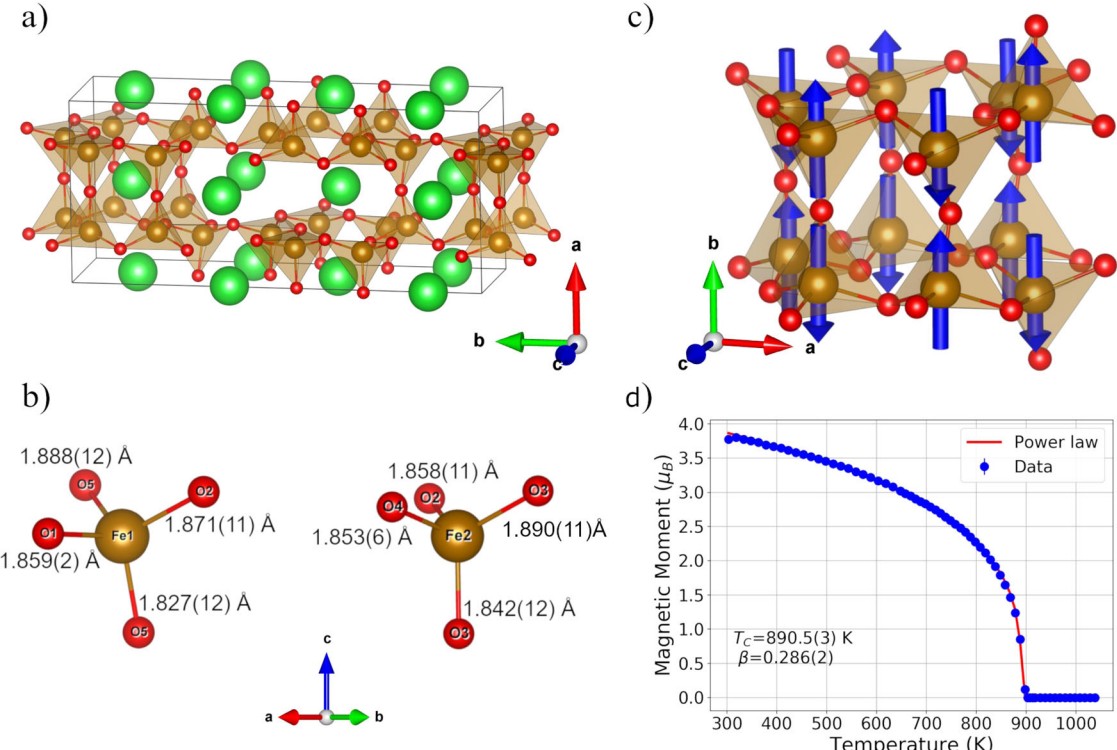

**Fig. 1 | γ-BaFe₂O₄ crystal and magnetic structures. a** Projection of the γ-BaFe₂O₄ nuclear structure obtained from the single crystal X-ray data. Green spheres represent the barium ions whereas orange and red spheres represent iron and oxygen, respectively. The projection shows the UUUUUD pattern of FeO₄ tetrahedra within the triangular cavities running along the [100] direction. **b** Coordination polyhedra for the two independent Fe positions. The average Fe−O bond lengths are 1.861(9) and 1.861(10) Å for Fe1 and Fe2 respectively, compatible with high-spin trivalent iron ions in tetrahedral coordination. **c** Magnetic structure of γ-BaFe₂O₄ with the G-type arrangement of the Fe³⁺ moments in the UUUUUD pattern of tetrahedra. This unit is coupled antiferromagnetically with the first neighbor ones. **d** Temperature evolution of the refined magnetic moment, in red is reported the fitting of the data with a critical law which returns a $T_N = 890.5(3)$ K and a β exponent of 0.285(2).

levels of the diffraction techniques, which is backed by the very low value of residual magnetization. As an example, if we consider all the residual magnetization of 0.0048 emu g⁻¹ in the 300 K loops to be due to the presence of barium hexaferrite ($M_r$ = 32 emu g⁻¹)[36] this leads to only a 0.15% in weight of the latter, well below the sensitivity of the diffraction measurements. At the same time, if we consider the contribution in the 800 K loop to be related to BaFe₄O₇, this will lead to a 0.16% in weight of the latter, again below the sensitivity of our diffraction measurement. The observation of still a very weak ferromagnetic component above 850 K (see Fig. S6) suggests the possible presence of further Fe oxides impurities also in this case likely below the 0.1% in weight.

Further confirmation of the lack of a weak ferromagnetic component in pure BaFe₂O₄ comes from the magnetic measurements performed on a thin film grown by pulsed electron deposition over a silicon substrate covered by a thin layer of thermal SiO₂. X-ray diffraction, Raman, and EDX measurements (see SI Figs. S7−9) were carried out to determine the quality of the sample, which appears to be a single phase γ-BaFe₂O₄ film displaying strong [021] orientation of the crystallites. The magnetization versus field data reported in Fig. 2c clearly indicates a purely antiferromagnetic behavior, in agreement with the results of the neutron diffraction investigations and symmetry analysis. Within this framework, NPD should be considered a phase-selective technique, in contrast with magnetometric characterizations that, despite higher sensitivity, intrinsically result in bulk-averaged data. Plots of the substrate and film raw data are reported in S10.

Bulk polycrystalline samples of γ-Ba₂FeO₄ were tested at room temperature through the Positive Negative (PN) protocol[38], described in detail in the Experimental section and in the SI, to verify the existence of a switchable electrical polarization, allowed by the polar point group of the structure. The quality of the ceramic pellet was analyzed by Rietveld method applied to PXRD data (Fig. S11), showing good agreement with the nuclear crystal structure model proposed by single crystal X-ray and neutron powder diffraction and absence of sizeable spurious phases. Figure 2d indicates that BaFe₂O₄ behaves as a ferroelectric ceramic, displaying a hard-like polarization hysteresis loop, with a coercive field $E_c$ = 12 kV cm⁻¹, and a remnant polarization $P_r$ = 0.18 μC cm⁻². The intrinsic nature of the polarization is also demonstrated by the correspondent integrated current during the measurement (Fig. 2d, inset) in which the polarization switching peaks (both positive and negative) are well defined. The observation of displacive current vs time (Fig. S13) confirms the retention of an overall capacitive behavior in the entire voltage range. Indeed, the current shows characteristic lobes with a maximum at an intermediate applied voltage value, on the rising part of the triangular pulses, and then a decrease until the voltage reaches the maximum amplitude. If leakage currents were the predominant contributions, a constant current increase should be observed instead of a sharp decrease since leakage is proportional to the applied voltage. Nonetheless, negligible dielectric losses are present near to the maximum amplitude (positive or negative) of the applied voltage as confirmed in the same regime by the tanδ plot reported in Fig. S14. The almost invariant I vs. V profile, during the applied voltage descent (i.e., second and fourth quarters, inset of Fig. 2d), further confirms the overall dielectric behavior of the compound. All the evidence indicate that despite the presence of negligible electric losses (tanδ < 0.06, Fig. S14), the observed electrical behavior of bulk γ-BaFe₂O₄ is related to the ferroelectric nature of the compound.

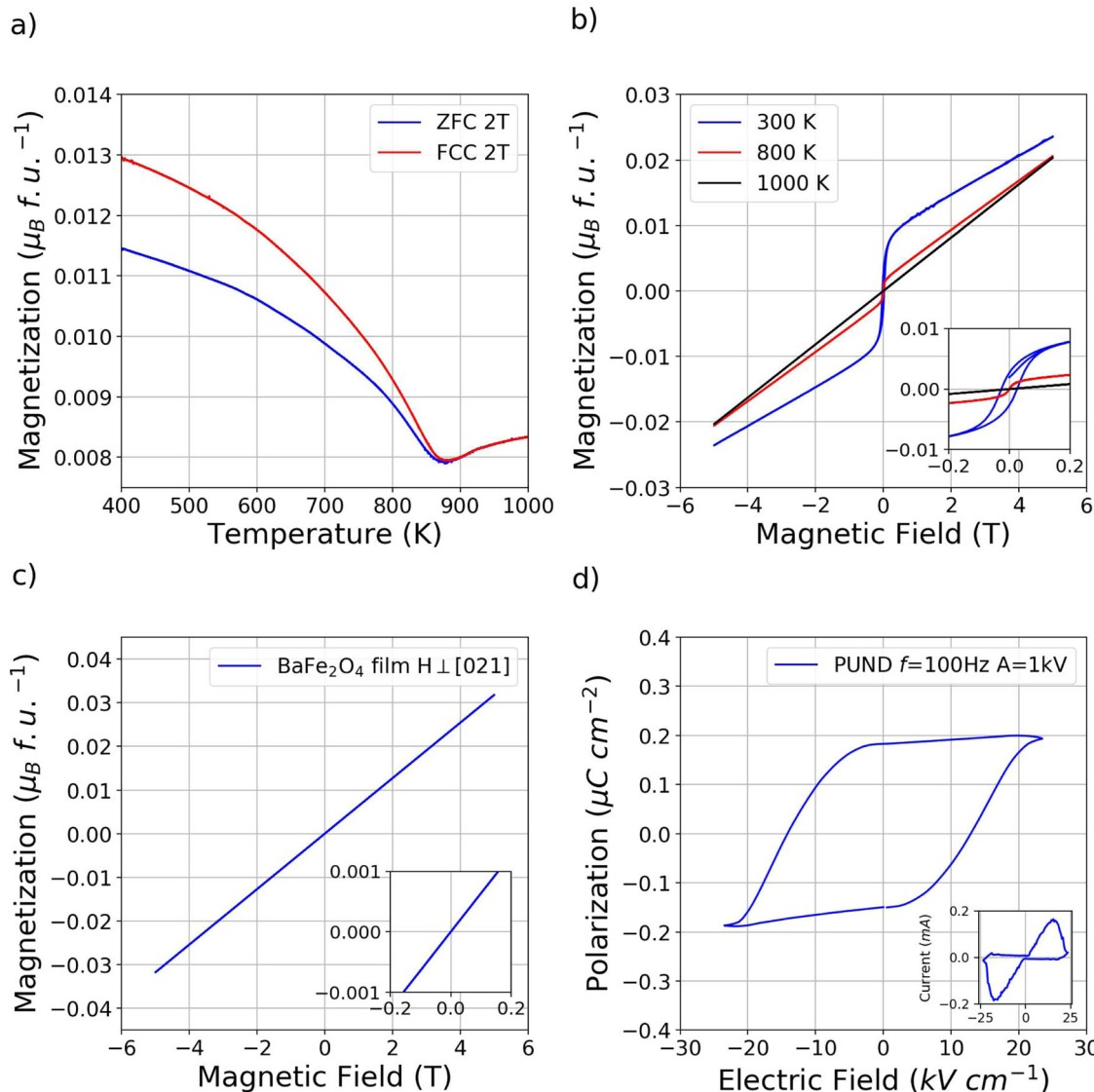

**Fig. 2 | γ-BaFe₂O₄ macroscopic measurements. a** ZFC and FCC magnetization measurements as a function of temperature collected at 2 T between 400 and 1000 K showing an anomaly at ~890 K in agreement with the neutron data. Please note the very small magnetic moment value in µB/f.u. excluding a ferro/ferrimagnetic transition. **b** Magnetic hysteresis loops measured at 300 K (blue curve), 800 K (red curve), and 1000 K (black curve) for a high purity sample of γ-BaFe₂O₄. The inset shows the low field regions of the loops showing the disappearing of the coercive field above 750 K indicating that is likely related to the presence of BaFe₁₂O₁₉ ($T_C$ = 720 K) as impurity. The remaining ferromagnetic component in the 800 K data is likely related to other Fe-containing impurities, e.g., BaFe₄O₇ ($T_N$ = 850 K). **c** Magnetic hysteresis loop collected at 320 K for a micrometric film of

γ-BaFe₂O₄ deposited by pulsed electron deposition on a monocrystalline silicon substrate covered by a thin layer of thermal SiO₂. The measurement was corrected by subtracting the diamagnetic signal of the substrate, additional information and plots of the raw data can be retrieved in the Supporting Information. The linear behavior without hysteresis and null coercive field agree with the neutron data which indicate an antiferromagnetic structure without ferromagnetic component. **d** PN electric polarization loop collected at a frequency $f$ = 100 Hz with triangular pulse of amplitude 1000 V. The inset shows the corresponding integrated current as a function of the applied voltage. The clear, symmetric current peaks reinforce the conclusion of a switchable polarization in the system.

The ferroelectric properties of γ-BaFe₂O₄ were further confirmed by piezoresponse force microscopy (PFM) measurements performed both on a sintered ceramic pellet (main text) and single crystals (Fig. S15). Figure 3 shows topography, lateral, and vertical PFM images of a relatively large grain of the ceramic sample. Distinct amplitude and phase contrast, especially for the lateral PFM image, correspond to ferroelectric domains with different polarization directions. These domains form a regular pattern with straight boundaries, which correspond to non-180° (90° or 120°) domain walls. It should be underlined that there are examples of ferroelectric materials where dominantly or only non-180 domains are observed[39]. Especially in ceramic materials these domains form to minimize the mechanical energy related to the spontaneous strain at the ferroelectric-to-

paraelectric phase transition and to minimize the mechanical stress arising at grain boundaries. In our specific case, there will be three types of elastic domains separated by 120° domain walls, which derive from the breaking of the 6-fold rotation axis of the parent structure[32] (as will be described in detail later), which were also observed in the single crystal diffraction measurements. The coexistence of several polarization variants (domains) is a strong proof in favor of ferroelectricity in the material studied.

To give further evidence of the ferroelectric character of the material, we also performed local polarization switching experiments. Figure 3g, f shows the local piezoresponse hysteresis loops measured on the same grain. The amplitude signal has a characteristic butterfly shape, typical of ferroelectric materials, and the phase of the vertical

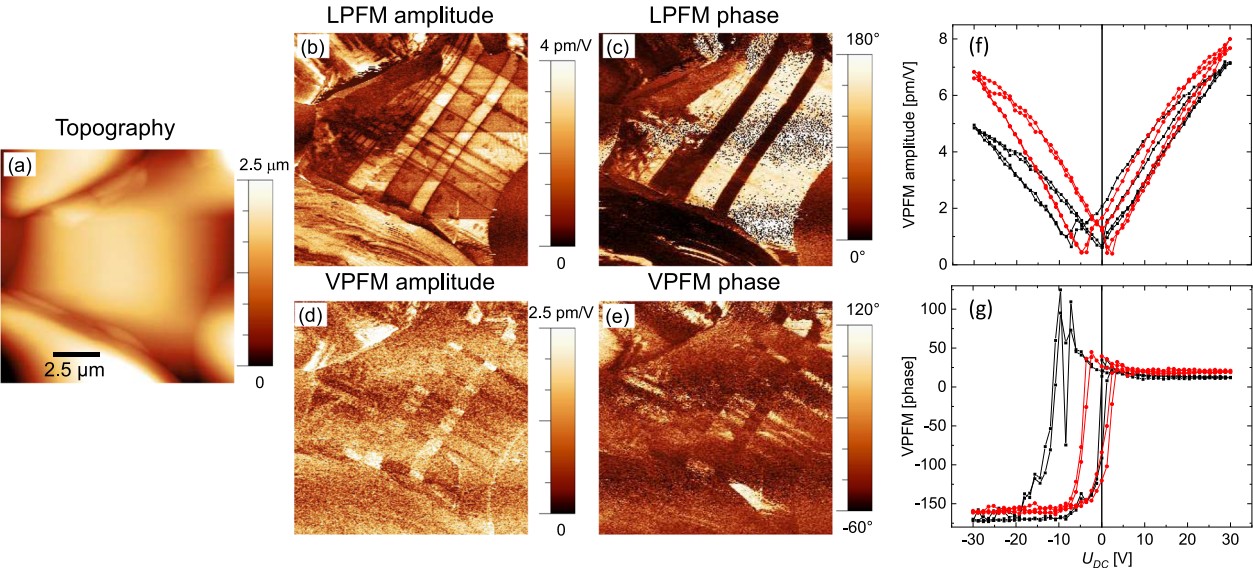

**Fig. 3 | PFM characterization of bulk γ-BaFe₂O₄ samples.** Topography (**a**), lateral PFM (**b**, **c**), and vertical PFM (**d**, **e**) images. **b**, **d** These show the amplitude of the PFM signals. **c**, **e** These show the phase of the PFM signals. Local piezoresponse hysteresis loops: amplitude (**f**) and phase (**g**). Black and red curves correspond to the first and the second switching cycles respectively.

PFM signal is changed by ~180°, which indicates switching between the states with the opposite direction of the out-of-plane polarization.

It should be noted that the PFM signal is relatively small, which correlates with the small value of the remanent polarization detected by PN measurements. Conversely, the coercive fields observed in the macroscopic polarization measurements and in the PFM hysteresis loop measurements are not quantitatively comparable. They have different physical meanings: the former corresponds to the situation when the volumes occupied by the domains with opposite polarities are equal, and the total macroscopic polarization becomes zero. The latter corresponds to the situation where the piezoelectric response from a newly created domain underneath the PFM tip is equal to the response from a still un-switched volume. Moreover, one has to take into account that by macroscopic measurements, a homogeneous electric field is applied to the sample, while in a PFM experiment the electric field created by the tip with the radius of 30 nm (in our case) is strongly inhomogeneous. Its value in the probed volume substantially exceeds the applied voltage divided by the sample thickness. Measurements carried out on single crystal samples obtained from the melt (Fig. S14) fully confirm the present findings.

## Discussion

The crystal structure of γ-BaFe₂O₄, with the *Cmc2₁* space group symmetry belonging to the polar *mm2* point group, allows the presence of an electrical dipole moment along the *c*-axis. PFM and P-E measurements carried out at room temperature (Fig. 2d) confirm the presence of a switchable polarization in the material indicating ferroelectric properties. The value of the saturation polarization is relatively small (~0.2 μC cm⁻²) if compared with proper ferroelectrics suggesting that the polarization in BaFe₂O₄ may be improper in origin. Symmetry analysis gives important information regarding the origin of the polarization modes and their coupling to the antipolar ones[40–43]. The high-temperature paraelectric phase of BaFe₂O₄ is reported to crystallize in the *P6₃22*[32] space group which does not have a group-subgroup relation with the *Cmc2₁* group determined in this work, suggesting that the transition is of the reconstructing type. In this case, it is likely that the two phases, high and low temperature, are both daughter phases of a common parent structure with respect to their possessing a group-subgroup relation. In the BaFe₂O₄ case both

paraelectric and ferroelectric phases can be described as daughter phases of the structure shown in Fig. 4a. This structure possesses the *P6₃/mmm* space group with a disorder of the apical oxygen positions and un-tilted oxygen tetrahedra around the Fe site and the *P6₃22* and *Cmc2₁* phases, as well as the other BaFe₂O₄ polymorphs[32], derive from different ordering patterns of the apical-O1 site and different rotation scheme of the oxygen tetrahedra. This approach of a disordered "latent" parent structure to describe reconstructive transitions between ordered phases has been used to describe the fcc-hcp transition in elemental metals and the graphite-diamond transitions[44].

The transition to the *Cmc2₁* structure can be described using two order parameters: $\xi$ related to the O2 position and to the in-plane rotation of the tetrahedra ($\Gamma_3^-$ irreducible representation (irrep) with propagation vector $k = (0,0,0)$, see Fig. 4a) and $\zeta$ which describes the O1 UUUUUD pattern and related atom displacements ($R_1$ irrep with propagation vector $k = (¼, 0, ½)$, see Fig. 4b). It is worth to underline that both these distortions if taken individually, do not break inversion symmetry and therefore are non-polar in nature. Nevertheless, their common action induces a displacive polar distortion described by the $\Gamma_6^-$ irrep with the order parameter $\rho$ (Fig. 4a). The latter distortion involves a polar displacement along the *c* direction of the orthorhombic cell, which is observed in the X-ray single crystal data, and results in the short Fe1–O5 and Fe2–O3 bonds (see Fig. 1b).

The free energy invariant, which describes the coupling between the oxygen distortions ($\xi$ for $\Gamma_3^-$ and $\zeta$ for R1) and the polar displacement ($\rho$), is $F_{coup} = \rho\xi\zeta^2$, and it gives important information about the polarization switching mechanism. Indeed, to reverse the sign of the spontaneous polarization $\rho$, the change in sign of the $\xi$ order parameter is required. On the contrary, since the $\zeta$ order parameter appears squared it does not require a change of sign. The polarization switching, therefore, entails the change of the in-plane rotation ($\Gamma_3^-$) leaving the ordering of the apical oxygen (O1) intact, as shown in Fig. 4c where the two polarization states are drawn. It is worth underlining that the transition from the parent structure to the *Cmc2₁* phase will generate three elastic domains with 120° boundaries, deriving from the breaking of the sixfold rotation axis, as observed by single crystal diffraction and by the PFM measurement.

To confirm the polar structure of γ-BaFe₂O₄ and the coupling term, ab-initio simulations based on density functional theory (DFT)

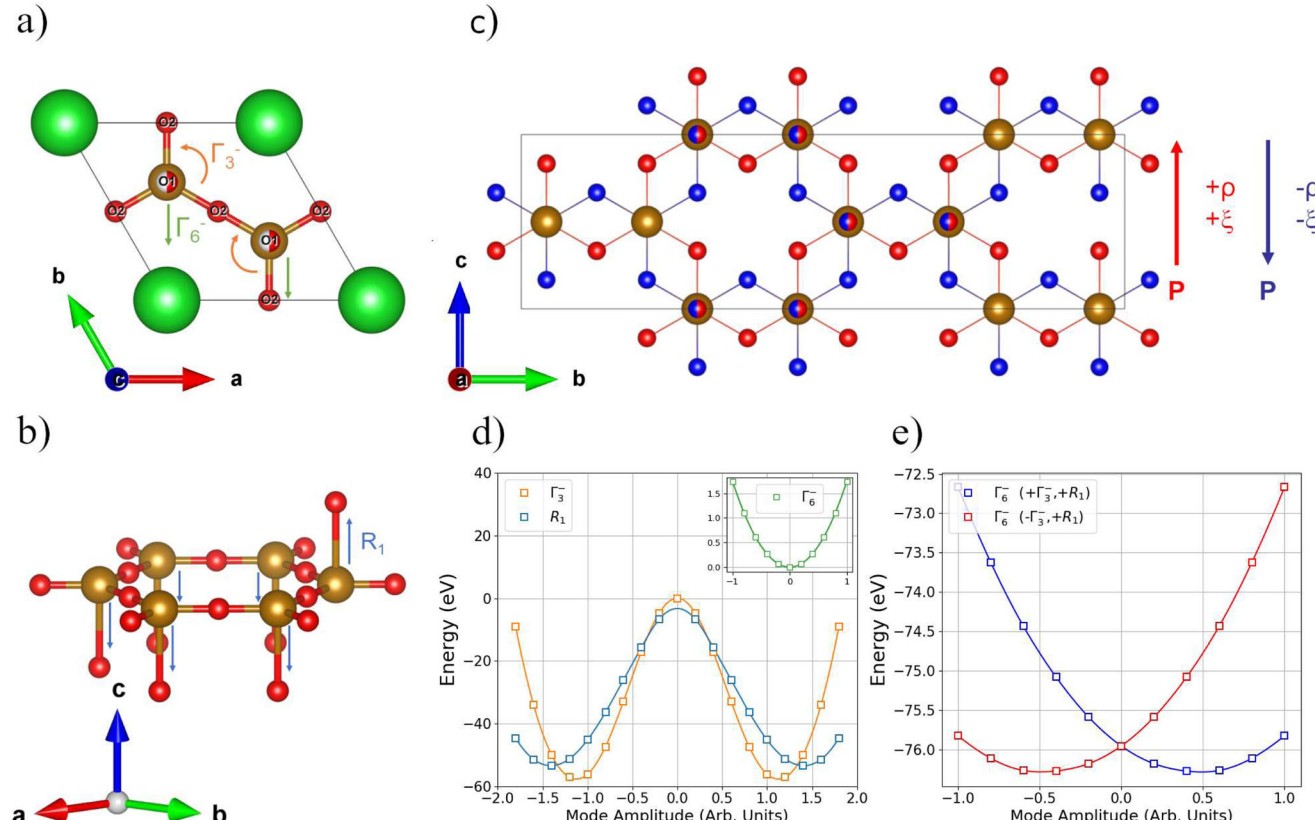

**Fig. 4 | Polarization switching mechanism in γ-BaFe₂O₄.** **a** Disordered "latent" parent structure of BaFe₂O₄ described in the $P6_3/mmm$ space group. The orange and green arrows denote the displacements due to the $\Gamma_3^-$ and $\Gamma_6^-$ modes. **b** O1 ordering and relative atoms displacements due to the R1 symmetry modes (blue lines). **c** Up (red oxygen positions) and down (blue oxygen positions) polarization states related by the rotation of the basal O2 positions as described by the $F_{coup}$ invariant in the text. **d** DFT total energies as a function of mode amplitudes, considering only the $\Gamma_3^-$ (orange line) and R1 (blue line) modes. The inset shows the total energy behavior, when activating only the $\Gamma_6^-$ mode. **e** DFT total energies obtained by changing the $\Gamma_6^-$ amplitude, in the presence of the R1 mode amplitude frozen to its positive minimum and the $\Gamma_6^-$ mode frozen to the positive minimum (blue points) and to the negative minimum (red points).

were performed. The structural model obtained from X-ray single crystal and neutron powder data was confirmed by structural relaxation calculations, starting from the parent structure shown in Fig. 4 (see Table S4 for a mode decomposition analysis[45]). The spontaneous polarization, calculated both with the Berry phase and point charge model (see Fig. S16), was evaluated to be ≈0.75 μC/cm², in good agreement with macroscopic measurements (it is worth to stress that the loop in Fig. 2d is performed on a powder sample). In order to prove the origin of the spontaneous polarization, we calculated the total energies of various distorted structures by changing the amplitude of the symmetry-adapted modes[45] calculated with respect to the $P6/mmm$ parent structure (see Table S4). The total energy profiles of the system, obtained by activating only the $\Gamma_3^-$ or only the R1 modes, are reported in Fig. 4d and show two symmetric minima, indicating that these two are soft modes for the system and can be considered the primary order parameters of the transition. On the contrary, the energy profile of the $\Gamma_6^-$ mode when considered as the only present distortion, shows a minimum at zero, hence being a hard mode for the system (Fig. 4d inset). The observation that the $\Gamma_3^-$ and R1 modes are soft modes for the systems, whereas the $\Gamma_6^-$ polar mode is a hard one, are consistent with the improper origin of the polarization in γ-BaFe₂O₄. To further confirm the improper character of the polarization and the coupling mechanism to the primary modes, we calculated the system total energy when varying the $\Gamma_6^-$ mode amplitude in the presence of the $\Gamma_3^-$ and R1 mode frozen at their positive minimum values. The energy profile is reported in Fig. 4e and it shows an asymmetric minimum at a positive value, confirming the improper / induced nature of the polar mode. The red curve in Fig. 4e shows the same energy

calculations with the $\Gamma_3^-$ mode time frozen to the negative minimum and the R1 still at the positive one. In this configuration the $\Gamma_6^-$ mode shows an asymmetric minimum at a negative value, confirming the coupling term which requires only the change of sign of the $\Gamma_3^-$ order parameter to switch the induced polarization. We remark that the bilinear–quadratic coupling of γ-BaFe₂O₄ strongly resembles the hybrid improper ferroelectric (HIF) mechanism[41,46] observed in perovskite-based materials, in which two non-polar distortions (octahedral tilting) give rise to a spontaneous polarization. Contrary to the present case in the HIF mechanism, the free energy coupling is trilinear, and the switching mechanism requires the change of either of the two octahedral tilting distortions making the switching mechanism more complex than the present case.

The high-temperature ferroelectric ($T_C > 1038$ K) and magnetic ($T_N = 890$ K) transitions of γ-BaFe₂O₄ make it a room-temperature multiferroic without the involvement of Bi or Pb atoms. Indeed, it possesses all the symmetry requirements to induce a multiferroic state as it happens in the gold standard BiFeO₃. The magnetic room temperature phase of γ-BaFe₂O₄ is a commensurate G-type structure without a weak FM moment because the propagation vector is at the Y point of the Brillouin zone. Nevertheless, as we show in the SI, it is possible to construct a Lifshitz invariant similar to the BiFeO₃ one, which indicates an instability in the system towards an incommensurate cycloidal state[47,48]. The reason why the γ-BaFe₂O₄ ground state is the commensurate G-type structure and not the incommensurate state is related to the different balance between competing interactions in particular between the Lifshitz invariant, single-ion anisotropy, and the staggered Dzyaloshinskii–Moriya interaction[49–51].

The results and analysis reported in this work clearly indicate that γ-BaFe$_2$O$_4$, and, more generally, the stuffed tridymite structures, host interesting multiferroic, and multifunctional properties. Indeed, our work shows, through a multi-technique, symmetry analysis and first-principles DFT-based approaches, that γ-BaFe$_2$O$_4$ is an anti-ferromagnetic improper ferroelectric with very high critical temperatures. The combination of crystallographic and macroscopic measurement techniques, combined with ab-initio calculations, allowed us to overcome each single-technique limitation and show the presence of a switchable polarization. The similarities, as well as the differences with BiFeO$_3$, make γ-BaFe$_2$O$_4$ an interesting playground for the study of coupling between magnetism and ferroelectricity, in particular if one considers that many compositions are known to share the same parent crystal structure AB$_2$O$_4$ (with A = Pb, Sr, Ca, Ba and B = Al, Fe, Ga)[25–31], allowing in principle to operate chemical substitutions to tune the physical properties of the compound. Finally, the preliminary attempts to grow the material in thin film form produced good quality samples, suggesting the compound to be quite stable and suitable to develop devices.

## Methods

### Synthesis

Polycrystalline samples were obtained by conventional solid-state reaction. BaCO$_3$ and Fe$_2$O$_3$ were mixed in equimolar amounts, ground, and then fired at 1573 K for 6 h in air. Thorough mixing and grinding are needed in order to avoid the formation of BaFe$_{12}$O$_{19}$, hardly detectable by PXRD, whose ferromagnetic contribution is large with respect to the γ-BaFe$_2$O$_4$ signal. PXRD Rietveld analysis of the obtained powders is reported in Fig. S11.

Single crystals were obtained from the melt of a slightly iron-rich mixture (BaCO$_3$/Fe$_2$O$_3$ molar ratio = 0.45/0.55). The stoichiometry was tuned to move out of the congruent melting composition, thus allowing the formation of a liquid phase promoting crystallization. Powders were mixed and pelletized, placed into a platinum crucible then slowly heated (0.5° min$^{-1}$) to 1613 K. After 8 h of reaction time the system was cooled down to 1123 K at 2 K min$^{-1}$, then to RT. Crystals were removed from the solidified melt using HNO$_3$ 0.1 M. The cooling rate is pivotal to obtain a pure γ-BaFe$_2$O$_4$ phase. Indeed, quenching the sample from 1173 K leads to a phase pure β′-BaFe$_2$O$_4$ as shown in Fig. S17.

### Thin film growth

γ-BaFe$_2$O$_4$ film was grown by pulsed electron deposition (PED) technique, a physical high-energy process based on the far-from-equilibrium evaporation of a solid target. The high energy pulsed electron discharge occurs inside a metallic hollow cathode and it is driven by a high voltage towards the target. The electron flow hits onto the surface of the target material ablating the very first layers (<1 nm). The interaction leads to the fast sublimation and to the formation of a plasma plume of target material, which finally condenses on a substrate placed in front of the latter. Thanks to the huge power density (>5 × 10$^7$ W cm$^{-2}$) transferred to the target, rapid heating and fast evaporation area ignite in non-thermodynamic conditions. This is the main feature of the technique, able to supply a high kinetic energy to the ablated species and finally allowing the formation of particular crystal structures and the preservation of the target stoichiometry on the growing film[52]. A high-density target was obtained by pressing γ-BaFe$_2$O$_4$ powders into a pellet with diameter 2.5 cm, sintered at 1573 K for 6 h. The PED depositions have been performed in the Channel Spark configuration[53] onto 0.5 × 0.5 cm$^2$-wide silicon substrates at high temperature (about 700 °C) under an O$_2$ pressure of 3.0 × 10$^{-2}$ mbar (5.0 purity degree), by setting the accelerating voltage at 13 kV, frequency rate of 6 Hz and a deposition rate around 0.13 Å/pulse, for a final thickness of 140 nm. All the samples were annealed in the same deposition atmosphere for 60 min. Before deposition, all the substrates were cleaned in acetone, and 2-prophanol for 15 min each in an ultrasonic bath, finally dried under air flow.

### Single crystal diffraction

Data were collected using a Bruker Smart diffractometer equipped with a Smart Breeze CCD detector. Graphite monochromatized Mo K$_\alpha$ radiation (λ = 0.71073 Å) was used as the incident beam. Intensities were integrated using SAINT[54] taking into account the presence of six twin domains and data reduction was carried out using the TWINABS program. The structure was solved using SHELXT[55] and refined full matrix with SHELXL[56] making use of anisotropic displacement parameters for all atoms.

### X-rays powder diffraction

Measurements were carried out in Bragg-Brentano geometry on a Rigaku Smartlab XE diffractometer making use of Cu K$_\alpha$ wavelength (λ = 1.5406 Å). A Ni filter was used to suppress the K$_\beta$ contribution. 5.0° soller slits were used both on the incident and diffracted beam and data were collected using a HyPix3000 detector operating in X-rays fluorescence reduction mode due to the presence of iron in the sample. Measurements were performed in the 10–120° 2θ range with 0.01 step size, 0.5° min$^{-1}$ speed, acquiring in continuous 1D mode. Rietveld refinements were performed using the GSASII software[57].

### Neutron powder diffraction

Data were collected on the WISH diffractometer at the ISIS facility (UK)[58]. The sample was contained in a thin wall vanadium can, and the diffraction data were collected in the temperature range 300–1038 K in a standard RAL5 furnace. High statistic data were collected at 300, 918, and 1038 K, whereas shorter runs were collected every 10 K to follow the structure evolution. The neutron diffraction Rietveld refinements were performed with the help of the JANA2006 software[59] and group theoretical calculation were performed with the ISODISTORT software[60].

### Magnetic characterizations

Magnetic measurements were performed with a Quantum Design MPMS 3 SQUID magnetometer located at the material characterization laboratory at the ISIS facility (UK) equipped with an oven for measurement in the temperature range 400–1000 K. A small fragment of the sample pellet (8.97 mg) was used for the measurements and ZINCAR cement has been employed allowing for good thermal contact and conduction with the oven probe. DC ZFC, FCC, and FCW measurements were performed with selected applied field values in the range 0.01-2 T. Magnetization versus field measurements were performed with the same sample set up at temperatures of 300, 800, 860, 870, and 1000 K in the field range ±5 T.

Magnetic measurement vs. field on γ-BaFe$_2$O$_4$ thin film was performed on Quantum Design MPMS-XL 5 T SQUID magnetometer. Initially, the blank diamagnetic contribution of the substrate, constituted by 5 × 5 mm square of 1 mm thick silicon piece of a commercial wafer covered by a thin layer of thermal SiO$_2$, was measured. Then, 1-micron thick BaFe$_2$O$_4$ film was grown on the same silicon substrate and measured in the same field and temperature condition. The magnetic signal was then corrected by subtracting the substrate contribution as reported in SI (see Fig. S10); this allowed to discriminate the weak film signal and obtain the reported data. During the magnetometric characterization, the film was located orthogonal with respect to the SQUID coils, with the external magnetic field applied parallel to the film surface.

### Electrical characterization

Ferroelectric measurements were carried out by TF-Analyzer 2000E produced by AIXACCT GmbH. The disc-shaped samples were lapped on both surfaces and then metalized by RT-sputtering with

200–300 nm of gold. The gold layers were contacted with 100 μm diameter gold wires using Du Pont conductive silver-based paste. The measurements were performed following a Positive-Negative (PN) protocol (further details can be found in the Supporting Information, Fig. S12). The frequency range explored was 100–1000 Hz, while the maximum applied voltage amplitude was 1500 V. Triangular PN reading pulses of frequency $f$ = 100 Hz and amplitude $A$ = 1000 V allowed to directly collect the ferroelectric hysteresis loop in a single stage, after applying a trapezoidal write pulse with the same amplitude and rise-time and duration of 5 ms.

### Piezo force microscopy measurements

PFM measurements[61] were performed using an MFP-3D atomic force microscope (Asylum Research). A cantilever-tip assembly Multi75E-G (Budget Sensors) with the spring constant 3 N m$^{-1}$ and Pt/Cr conductive coating was used. The PFM images were taken at ac voltage with the amplitude 10 V and frequency 300 kHz applied to the tip. The chosen frequency was far from the resonance frequency for both vertical and lateral PFM signals. The local piezoresponse hysteresis loops were measured in dual amplitude resonance tracking PFM mode. A sequence of dc voltage pulses that magnitude steps in time was applied to the PFM tip. Between the voltage steps, the PFM response was measured by applying ac voltage with the amplitude of 3 V. The pulse duration and duty cycle were 50 ms. The PFM data were processed using the Gwyddion software.

### Raman spectroscopy

Measurements were carried out using a micro-Raman spectrometer (Horiba LabRam HR Evolution Raman) equipped with confocal Olympus microscope and 10×, 50×, ULWD50×, 100× objectives (spatial resolutions of ~1 μm). The Micro-Raman apparatus is completed by a He-Ne laser emitting at 632.8 nm, BraggRate Notch Filters, Silicon CCD + InGaAs Diode Array detectors, gratings 300–600–1800 lines/mm, and density filters. The spectrometer was calibrated using the standard silicon Raman peak at 520.6 cm$^{-1}$ before each measurement. The spectra reported here were recorded using the 100× objective, for 30 s and four repetitions, using a 5% density filter.

### Energy Dispersive X-ray Spectrometry

Compositional analysis was performed by collecting the Energy Dispersive X-ray Spectrometry spectrum at 25 keV, through a Scanning Electron Microscopy apparatus (SEM, Philips 515) EDAX probe (Phoenix with Si:Li detector).

### Calculation methods

First-principles DFT simulations were performed using the Vienna Ab initio Simulation Package (VASP)[62,63]. The generalized gradient approximation (GGA) based on the Perdew–Burke–Ernzerhof functional[64] was employed to treat the exchange–correlation interaction. The localized 3d electron correlation of Fe atoms was considered by using the GGA + U method[65,66], with a Hubbard U parameter chosen to be 4.0 eV[67]. The projector-augmented-wave potentials[68,69] were used to describe the electron-ion interaction. In all calculations, the lattice constants and the ground state magnetic configuration were used, as obtained experimentally. The energy cut off was selected to be 500 eV for the plane-wave basis set. The Brillouin zones were sampled using a $4 \times 2 \times 6$ k-grid mesh in the Γ-centered scheme. The forces convergence standard on each atom was chosen as 0.01 eV/Å and the self-consistent calculations were stopped when the energy difference was smaller than $10^{-6}$ eV per atom. The ferroelectric polarization was calculated using the Berry phase method within the modern polarization theory[70,71].

[Further details of the crystal structure investigation may be obtained from the Fachinformationszentrum Karlsruhe, 76344 Eggenstein-Leopoldshafen (Germany), on quoting the depository number CSD-2092798].

## Data availability

The data presented in the main text are available at https://doi.org/10.5281/zenodo.7400443. Raw data of the neutron experiment are available at https://doi.org/10.5286/ISIS.E.RB1820503. Single crystal structure factors are present in the .cif file uploaded as Supplementary Information. First-principles calculations were performed using the licensed VASP code[62,63]. The rest of the data are available from the corresponding authors upon reasonable request.

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

## Acknowledgements

F.O. thanks Dr. Pascal Manuel and Dr. Dmitry D. Khalyavin for fruitful discussions. Dr. Davide Calestani is acknowledged for EDX data collection. The authors thank the Science and Technology Facility Council for the neutron beam time allocation at the WISH diffractometer (ISIS) under the proposal https://doi.org/10.5286/ISIS.E.RB1820503. The authors would like to thank Dr. Gavin Stenning for help on the SQUID magnetometer measurement at the Materials Characterisation Laboratory at the ISIS Neutron and Muon Source. F.M. thanks Prof. Ludovico Cademartiri for the discussion. This work has benefited from the equipment and framework of the COMP-HUB Initiative, funded by the 'Departments of Excellence' program of the Italian Ministry for Education, University and Research (MIUR, 2018-2022).

## Author contributions

F.M. and M.C. synthesized the bulk and ceramic samples. F.M. and E.C. synthesized the single crystals. P.G., S.R., M.C., and E.G. deposited the thin film samples. F.M. and G.C. carried out the X-ray diffraction experiments and performed data analysis. F.O. performed the neutron diffraction experiments and analyzed the data. D.D. and F.O. performed the SQuID measurements; D.D. interpreted the magnetometric data and M.S. participated in the analysis. D.D. prepared the samples and carried out the bulk electric measurements and interpretation. V.V.S. performed piezo force microscopy. F.O. carried out the symmetry analysis and modes decomposition. C.L., W.R., and S.P. performed the DFT calculations. G.S. collected the Raman spectroscopy data. F.O., D.D., P.G., and F.M. discussed the results. F.M. supervised the research. All the authors contributed to the manuscript; writing was supervised by F.O. and F.M.

## Competing interests

The authors declare no competing interests.
