## [Peer Review File · Nature Communications]

γ -BaFe₂O₄: a fresh playground for room temperature multiferroicityREVIEWER COMMENTS

Reviewer #1 (Remarks to the Author):

REFEREE REPORT

on paper " γ -BaFe₂O₄: a fresh playground for room temperature multiferroicity" (NCOMMS-22-14709) by authors Fabio Orlandi, Davide Delmonte, Gianluca Calestani, Enrico Cavalli, Edmondo Gilioli, Vladimir V. Shvartsman, Patrizio Graziosi, Stefano Rampino, Giulia Spaggiari, Massimo Solzi, Michele Casappa, Francesco Mezzadri, submitted to Nature Communications

The paper " γ -BaFe₂O₄: a fresh playground for room temperature multiferroicity" is devoted to investigation of the complex Ba-ferrite (BaFe₂O₄) in single crystal form. The crystal and dual ferroic properties were investigated. Authors analyzed magnetic (magnetization as function of the temperature and external magnetic fields) and electrical properties (polarization as function of the external electrical field). Coexistence of the dual ferroic ordering opens broad perspectives for practical applications. Object attracts great attention due to scientific importance and practical interest. The work is of scientific interest to specialists in the field of obtaining magnetic materials for using as sensors, memory devices with double type of ferroic ordering. Authors provided detailed description of the multiferroic origin and discussed the correlation between the features of the local crystal structure, the peculiarities of the magnetic structure (Fe-O bonds etc.) and microscopic dual ferroic ordering. The problem of the features of the magnetic ordering in the complex oxides is very actual and important. The data are reliable and do not cause much doubt. Nevertheless, there are several points before the paper can be published. I hope that authors after revision can improve the paper and can published it in NatComm. My decision at this stage is minor revision. But I hope that after brutal revision it can be accepted. I impressed by the paper.

1. Abstract: I feel it is too short and non-informative. Please add any supporting information for statements that γ -BaFe₂O₄ is "as solid candidates for application". Please add most important results obtained in paper.
2. The choice of the research object is attractive. But authors discussed in Introduction only I-st type multiferroics (based on BiFeO₃). But there are a lot of other compounds based on ferrites that demonstrated double ferroic ordering (for example quasi-centrosymmetrical hexagonal ferrites based on Al- and In-substituted BaFe₁₂O₁₉). These compounds are also perspective candidates for room-temperature applications.
3. This is very interesting that authors concluded that crystal structure of the investigated single crystal can't be described in the framework of the centrosymmetric SG (Cmcm). The same results were obtained by other researchers for different Ba-ferrites (BaFe_{12-x}Al_xO₁₉, and BaFe_{11.9D0.1}O₁₉, where D= In, Sc and Ga). Please discuss this.
4. What is the oxygen stoichiometry of prepared samples? It is well known that the complex 3d-metal oxides easily allow the oxygen excess and/or deficit. Oxygen excess and deficit can change magnetic/electrical properties for the complex oxides. Did authors control oxygen content?
5. Authors demonstrate the features of the local crystal structure by analysis of the Fe-O bond length. But I propose for further investigation apply ab initio techniques (first principles total energy projector-augmented wave method) with the hybrid potential it was possible to identify minima potential relief for specific Fe ions, while the height of the energy barrier defined the type of the electric polarization. Good correlation of the neutron diffraction data and calculation of the energy state for different Fe ions can lead to explanation of the principal approaches that explain the origin of the multiferroicity.
6. This is not strong requirement, but I propose replace "Magnetization" (on X-axis Fig. 2a-c) by the "Magnetic moment". I feel that this is "effective magnetic moment" per formula unit in Bohr magneton.

Reviewer #2 (Remarks to the Author):

The contribution by Orlandi et al. reported a room temperature multiferroicity, γ -BaFe₂O₄. Finding functional materials with room temperature multiferroicity is always an intriguing task because of its

considerable challenge. The reported results convinced me that γ -BaFe₂O₄ does have a temperature multiferroicity. However, some issues should be fully stressed before considering acceptance.

1. The temperature-dependent magnetization shown in Fig. 2a and Fig. S5a-b are from the same sample. Why a higher magnetic field of 2T in Fig. 2a gives a weaker magnetization than 0.1 T in Fig. S5b?
2. I think it would be better if the authors can show the raw data for Fig. 2c in the Supporting Information, and demonstrate how the diamagnetic background was subtracted.
3. The thin film grown by the pulsed electron deposition in Fig. 2c shows a purely antiferromagnetic behavior. Why the authors did not perform other magnetic measurements based on this sample? I think the pure phase could give a better argument for this work.
4. In Figure 3, the authors labeled (f) and (g) in the caption by mistake.
5. In Figure 3(f, g), what do the black and red loops represent?
6. Since the ferroelectric domains can be well observed by PFM, it is necessary to conduct magnetic force microscopy (MFM) measurements to unveil the magnetic domains.
7. More evidence should be provided to reveal the crystal structure, such as STEM.

Reviewer #3 (Remarks to the Author):

This manuscript reports the detailed crystal and magnetic structure analysis of γ -BaFe₂O₄. It is clearly shown that ferroelectric and antiferromagnetic orderings coexist in this compound at room temperature. The presence of ferroelectricity is further confirmed by P-E and PFM measurements. The obtained experimental data and analyzed structural parameters are quite reliable establishing the title compound a new room temperature multiferroic compound. However, on the other hand, the spontaneous electric polarization is tiny, only 0.2 $\mu\text{C}/\text{cm}^2$ and no spontaneous magnetic moment owing to the spin canting is present. In addition, no correlation between the ferroelectricity and magnetism is proposed. Moreover, the polar nature is not a new finding, but the space group is the same as reported in the previous report (ref. 30). Judging from these facts, the impact to the field is limited and will not attract broad interest of the readers of Nature Communications. I find it should appear in a more specialized journal.

REFEREE #1

The paper “ γ -BaFe₂O₄: a fresh playground for room temperature multiferroicity” is devoted to investigation of the complex Ba-ferrite (BaFe₂O₄) in single crystal form. The crystal and dual ferroic properties were investigated. Authors analyzed magnetic (magnetization as function of the temperature and external magnetic fields) and electrical properties (polarization as function of the external electrical field). Coexistence of the dual ferroic ordering opens broad perspectives for practical applications. Object attracts great attention due to scientific importance and practical interest. The work is of scientific interest to specialists in the field of obtaining magnetic materials for using as sensors, memory devices with double type of ferroic ordering. Authors provided detailed description of the multiferroic origin and discussed the correlation between the features of the local crystal structure, the peculiarities of the magnetic structure (Fe-O bonds etc.) and microscopic dual ferroic ordering. The problem of the features of the magnetic ordering in the complex oxides is very actual and important. The data are reliable and do not cause much doubt. Nevertheless, there are several points before the paper can be published. I hope that authors after revision can improve the paper and can published it in NatComm. My decision at this stage is minor revision. But I hope that after brutal revision it can be accepted. I impressed by the paper.

RE: We thank the referee for the very positive review of our work and for his pertinent comments which improved our manuscript.

1. Abstract: I feel it is too short and non-informative. Please add any supporting information for statements that γ -BaFe₂O₄ is “as solid candidates for application”. Please add most important results obtained in paper.

We have updated the abstract with the most relevant results of the present work. Considered the 150 words limit imposed by the Journal we have chosen to soften the statement, which is properly addressed in the main text.

2. The choice of the research object is attractive. But authors discussed in Introduction only I-st type multiferroics (based on BiFeO₃). But there are a lot of other compounds based on ferrites that demonstrated double ferroic ordering (for example quasi-centrosymmetrical hexagonal ferrites based on Al- and In-substituted BaFe₂O₄). These compounds are also perspective candidates for room-temperature applications.

Some examples of different room temperature multiferroic systems were already present (references 12-16), however we thank the Referee for the suggestion: a couple of examples of the hexaferrite-derived multiferroics have now been added to the introduction.

3. This is very interesting that authors concluded that crystal structure of the investigated single crystal can't be described in the framework of the centrosymmetric SG (Cmcm). The same results were obtained by other researchers for different Ba-ferrites (BaFe₁₂AlO₁₉, and BaFe_{11.9D0.1O19}, where D= In, Sc and Ga). Please discuss this.

We agree that a symmetry-comparative approach can be particularly informative in specific ambiguous structures. A statement regarding similar cases (SrFe_{10.8}In_{1.2}O₁₉, BaFe_{12-x}Al_xO₁₉, BaFe₄O₇) has been added to the main text together with the corresponding citations.

4. What is the oxygen stoichiometry of prepared samples? It is well known that the complex 3d-metal oxides easily allow the oxygen excess and/or deficit. Oxygen excess and deficit can change magnetic/electrical properties for the complex oxides. Did authors control oxygen content?

RE: We thank the referee for the pertinent comment. The oxygen content, as well as the other cations stoichiometry, has been checked during the refinement of the neutron and x-ray diffraction data. The x-ray measurements gave us good information about the heavy atoms in the system whereas neutron diffraction about the oxygen content. Neutrons are in fact very sensitive to the light atoms and the refinement indicates that all the atomic positions are fully occupied, within the resolution of the diffraction measurement, in agreement with the BaFe_2O_4 stoichiometry.

We added a sentence in the page 5 of the manuscript to highlight this point.

5. Authors demonstrate the features of the local crystal structure by analysis of the Fe-O bond length. But I propose for further investigation apply ab initio techniques (first principles total energy projector-augmented wave method) with the hybrid potential it was possible to identify minima potential relief for specific Fe ions, while the height of the energy barrier defined the type of the electric polarization. Good correlation of the neutron diffraction data and calculation of the energy state for different Fe ions can lead to explanation of the principal approaches that explain the origin of the multiferroicity.

RE: We thank the referee for the pertinent suggestion. We have indeed investigated this system by DFT calculations, and these results are now reported in the main text as well as in the supplementary materials.

We first confirmed the observed nuclear structure by relaxing the parent structure shown in the manuscript and we obtained good agreement as can be seen from the comparison of the mode decomposition of the observed and relaxed structures reported in table S4. We also confirmed the presence of a spontaneous polarization in the material with the same order of magnitude as the one observed in the macroscopic measurements.

Unfortunately, the determination of the exact energy barrier for the polarization switching was not possible due to the uncertainties related to the real switching path. Nevertheless, we provide a study of the coupling mechanism between the different distortions confirming the coupling term derived on symmetry basis.

We first performed a mode decomposition analysis of the relaxed structure with respect to the parent $P6/mmm$ structure reported in the main text. This mode decomposition allows us to separate all the distortions that owns the same symmetry and, in this way, we can calculate the system energy by turning on or off a specific distortion mode.

We then showed that the Γ_3 - mode, which represents the rotation of the oxygen tetrahedra, and the R1 mode, which envisage the UUUUD ordering of the apical oxygens and displacement of the other atoms with the same pattern, are soft modes of the system if are taken as the only active modes in the structure. The energy calculation (figure1, right panel) indeed shows two symmetric minima at a finite value of the mode amplitude. The DFT calculation suggests that these two modes are primary order parameters for the system. Moreover, the DFT calculation shows that if we take the polar Γ_6 - mode as the only active mode this is a hard mode for the system showing a minimum of the energy at zero amplitude.

In order to confirm the improper nature of the polarization we performed energy calculations by varying the amplitude of the polar Γ_6^- mode while we froze the value of the Γ_3^- and R1 modes to their positive equilibrium amplitude values. In this case the Γ_6^- mode amplitude shows only one minimum at a finite positive value confirming the improper nature of the polarization and confirming also the observation that in order to obtain a finite polarization it is necessary the presence of both the Γ_3^- and R1 distortions. Finally, we confirmed the bilinear-quadratic coupling term between the three distortions by calculating the energy of the Γ_6^- mode while the R1 mode is frozen to the same positive value but the Γ_3^- mode is now frozen to the negative equilibrium value. In this case, the energy profile of the Γ_6^- mode shows only one minimum but now at a negative value (red curve in the right side of the figure below). This calculation confirms the switching mechanism, which requires the switching of the tetrahedra rotation (Γ_3^-) to change the direction of the polarization, discussed in the text based on symmetry considerations.

Figure 1 (left) DFT total energies as a function of mode amplitudes, considering only the Γ_3^- (orange line) and R1 (blue line) modes. The inset shows the total energy behaviour, when activating only the Γ_6^- mode. (right) DFT total energies obtained by changing the Γ_6^- amplitude, in presence of the R1 mode amplitude frozen to its positive minimum and the Γ_6^- mode frozen to the positive minimum (blue points) and to the negative minimum (red points).

We added all these results to the discussion part of the manuscript as well as in the supplementary materials. We think that these calculations confirm our understanding of the material and reinforce our conclusion of a new coupling term for improper ferroelectricity.

We would like to thank the referee again for suggesting this investigation which has definitely improved our manuscript.

6. This is not strong requirement, but I propose replace “Magnetization” (on X-axis Fig. 2a-c) by the “Magnetic moment”. I feel that this is “effective magnetic moment” per formula unit in Bohr magneton.

RE: We thank the referee for the comment but in this instance, we think that magnetization is more appropriate since it is the macroscopic property that the MPMS measures. To the best of our knowledge, the terminology “effective magnetic moment” is usually used for estimating the magnitude of intrinsic magnetic character of a material as obtained via Curie-Weiss interpolation within its paramagnetic state. We have then provided these values in terms of

uB/f.u. to underline that the value of the magnetization is extremely small and likely to be related to impurities.

Reviewer #2 (Remarks to the Author):

The contribution by Orlandi et al. reported a room temperature multiferroicity, γ -BaFe₂O₄. Finding functional materials with room temperature multiferroicity is always an intriguing task because of its considerable challenge. The reported results convinced me that γ -BaFe₂O₄ does have a temperature multiferroicity. However, some issues should be fully stressed before considering acceptance.

We thank the referee for their very positive evaluation of our work and for the comments that improved our manuscript.

1. The temperature-dependent magnetization shown in Fig. 2a and Fig. S5a-b are from the same sample. Why a higher magnetic field of 2T in Fig. 2a gives a weaker magnetization than 0.1 T in Fig. S5b?

RE: We would like to thank the referee for pointing this inconsistency out. There has been a mistake on our side and the data shown in figure S5b refer to a different sample batch with respect to the one presented in the main text and in figure S5a. The sample of figure S5b contains a much higher content of barium hexaferrite and it is the same sample batch giving rise to the loop shown in figure S5c.

The FCC and FCW curves measured on the same sample shown in the main text are reported below and the magnetization values are consistent with the remaining measurements shown in the work.

We changed panel S5b with the above measurement in the revised SI and we apology for our fault.

2. I think it would be better if the authors can show the raw data for Fig. 2c in the Supporting Information, and demonstrate how the diamagnetic background was subtracted.

We thank the Referee for asking more details on the experimental/analytical procedure we followed to extract the magnetic signal of the BFO film from the magnetic measurement of the entire sample, in which almost all the mass is represented by the diamagnetic Si substrate.

To deal with this, we defined a three-step experimental protocol: 1) the substrate (constituted by a cut of Si wafer covered with a thin layer of thermal SiO₂) was measured cycling from 5T to -5T with a standard hysteresis loop SQUID protocol; (2) the same measurement protocol was performed on the same substrate after the growth via PED of the BFO film; (3) the raw data (emu/g) were point-by-point subtracted assuming $M_{\text{film}} = M_{\text{blank+film}} - M_{\text{blank}}$, and then plotted scaling the unit to $\mu\text{b}/\text{F.U.}$ in Fig. 2c of the manuscript, for consistency with the other magnetic measurements of the paper.

The measurements (1) and (2) are now reported in figure S10, while the subtraction is, as said, reported as Figure 2c.

As it can be appreciated here, the substrate signal (black curve, Fig. S10a) is characterized by an almost ideal diamagnetic response, as expected for Si and an intense negative slope. In comparison, the sample constituted by substrate and thin film shows a diamagnetic susceptibility significantly reduced due to the presence of a superimposed positive trend of the magnetization coming from BFO (blue curve in Fig. S10a). Slight deviation from the linearity is observed in near zero-field conditions, where probably traces of ferromagnetic components, already observed in the bulk at higher concentrations, contribute to flatten the negative slope of the diamagnetic blank almost completely (Fig. S10b)

This description was added in the main text of SI as a comment of this experimental result, while a short sentence was added also in the main paper in the Methods Section.

3. The thin film grown by the pulsed electron deposition in Fig. 2c shows a purely

antiferromagnetic behavior. Why the authors did not perform other magnetic measurements based on this sample? I think the pure phase could give a better argument for this work.

We thank the referee for the pertinent comment. We have now tried to perform magnetization versus temperature measurements also for the thin film samples. Unfortunately, from our attempts on films of comparable thickness with respect to the one reported in the main text (150 nm) it seems that the samples experience a sharp change in the first derivative around 950 K as can be seen, as example, from the ZFC curve shown in the plot below where the magnetization shows a sharp rise. This measurement is performed on the as-grown sample without any previous thermal history. Since it was the first time we observed such anomaly at 950 K, we argued that this phenomenology is related to something occurring on the film composition, probably due to extrinsic reasons. The sample was put in the SQUID oven probe in direct contact with the alumina support. Noteworthy, the film appearance changes after extraction from the instrument, so we are speculating that some chemical interactions occurred, activated by the high temperature conditions and resulting in some kind of contamination. We are performing characterizations to confirm this point, however we would like to stress that it is unlikely a decomposition process to occur, as no effect was observed at the same temperatures during neutron diffraction experiments.

The sharp state change is observed in the field cooled measurement as well and it can be seen from the figure below that the magnetization is increased by around 30 times from the ZFC low temperature trend. We would like to stress that we performed the measurements with the same protocol on various film samples of different thicknesses observing similar effects in all cases. We are developing an updated measurement setup to conduct high temperature magnetic measurements on the thin films, however strong constraints derive from the need to fix the sample to the probe without the use of polymeric materials.

In any case, if we look at the same ZFC measurement, performed on the as grown sample, and we cut the curve below 850 K (see following image), we are reasonably looking into the intrinsic AFM BaFe₂O₄ film response; indeed, the thermal trend is here qualitatively comparable to what observed and plotted in Fig 2a of the main text for the bulk sample, with a minimum at 800 K characteristic of a fully-ordered AFM structure. At lower temperature the weak FM component becomes predominant, despite a seemingly reduced increasing trend of the magnetization down to 400 K. Consequently, and consistently to what reported in the paper, we conclude that the weak FM signal is to be ascribed to extrinsic phases, in this case extremely reduced with respect to the bulk samples.

4. In Figure 3, the authors labelled (f) and (g) in the caption by mistake.

RE: We thank the referee for highlighting the typo which has been correct in the new version of the manuscript

5. In Figure 3(f, g), what do the black and red loops represent?

The two curves represent the first and the second switching cycles. The label has been updated in the new version of the manuscript.

6. Since the ferroelectric domains can be well observed by PFM, it is necessary to conduct magnetic force microscopy (MFM) measurements to unveil the magnetic domains.

We thank the referee for the suggestion. The main purpose of the PFM measurements was to confirm the ferroelectric state of γ -BaFe₂O₄ and we were able to observe ferroelectric domains and to demonstrate local polarization switching. The studied material is a pure antiferromagnet without spin canting, which is confirmed by neutron diffraction measurements. Due to this, the MFM signal that we would be able to observe will be localized only on the domains boundary and will be present only if the antiferromagnetic domain walls possess an uncompensated ferromagnetic moment.

For these reasons, the observation of the antiferromagnetic domain walls by MFM at room temperature is a difficult task. Most AFM domain walls do not carry uncompensated moments, or the moment size is too small to be detected by typical magnetic imaging techniques. [Cheong, SW., Fiebig, M., Wu, W. et al. Seeing is believing: visualization of antiferromagnetic domains. *npj Quantum Mater.* 5, 3 (2020). <https://doi.org/10.1038/s41535-019-0204-x>]. In literature there are only a few reliable reports on MFM imaging of antiferromagnetic domains, all of them were done at cryogenic temperatures. Of course, this can be an interesting project and, in the future, we will conduct MFM research on antiferromagnetic domains in BaFe₂O₄.

7. More evidence should be provided to reveal the crystal structure, such as STEM.

RE: In our opinion the information provided by a STEM experiment would not allow a significant advance in the knowledge of the crystal structure of the system. Indeed, high-quality

x-ray single crystal data and state of the art powder neutron diffraction data give independently the same results within a 0.03 Å uncertainty on the bond distances, while the resolution of electron microscopy techniques is well beyond this value. In order to have a completely different point of view about the crystal structure of BaFe₂O₄, we performed instead DFT ab-initio calculations to provide an energy-related information. Structure relaxation starting from the parent P6/mmm phase, shown now in figure 4a, has been indeed performed revealing an energy minimum in excellent agreement with the structure as determined from diffraction data, as shown by comparison of the mode decompositions in table S4.

Conversely, if the Referee's suggestion is to fully unveil the (probably rich) microstructure of this material, we agree that electron microscopy will be a fundamental tool and we are planning to perform specific measurements in the future, but we believe such task would go beyond the scope of the present paper.

Finally, we would like to underline that the obtained crystal structure is now supported by two different diffraction techniques, by group theory and DFT calculations and our interpretation allows to thoroughly explain all the observed macroscopic properties.

Reviewer #3 (Remarks to the Author):

This manuscript reports the detailed crystal and magnetic structure analysis of g-BaFe₂O₄. It is clearly shown that ferroelectric and antiferromagnetic orderings coexist in this compound at room temperature. The presence of ferroelectricity is further confirmed by P-E and PFM measurements. The obtained experimental data and analyzed structural parameters are quite reliable establishing the title compound a new room temperature multiferroic compound. However, on the other hand, the spontaneous electric polarization is tiny, only 0.2 uC/cm² and no spontaneous magnetic moment owing to the spin canting is present. In addition, no correlation between the ferroelectricity and magnetism is proposed. Moreover, the polar nature is not a new finding, but the space group is the same as reported in the previous report (ref. 30). Judging from these facts, the impact to the field is limited and will not attract broad interest of the readers of Nature Communications. I find it should appear in a more specialized journal.

RE: We are sorry to see that the novelty of our work was not appreciated by the Referee and we apology if the manuscript was not clear enough in highlighting some of the novelties our work brings to the field. On the other hand, we appreciate the quality of the experimental data and analysis is recognized, and the fact that the Referee considers our conclusions reliable.

We agree that the electrical polarization is small, and that the system does not show a spontaneous ferromagnetic moment. Nevertheless, the origin of the improper polarization and the symmetry characteristic of the magnetic ordering and of the system more in general make γ-BaFe₂O₄ and the stuffed tridymite material a good playground for RT multiferroicity, and we hope our work will spur the research community efforts on these materials. Indeed, as we highlight, the symmetry analysis we performed clearly indicate the improper nature of the spontaneous polarization linked to the UUUUD pattern of the FeO₄ tetrahedra. To the best of our knowledge the bilinear-quadratic coupling term derived in the manuscript has never been reported in the improper ferroelectricity field and this can give a new spur to the community beyond the hybrid improper mechanism observed in the layered perovskites materials. In the revised manuscript, we better highlight these points, and we also provide an ab-initio investigation, using DFT methods, to prove that the bilinear-quadratic term is indeed the origin of the improper polarization in these compounds.

In addition, it is true that the polar symmetry of the compound was already known but “polar” does not mean “ferroelectric” as many piezo- or pyro-electric compounds are known, being characterized by a polar symmetry but non-switchable domains. In the present case, we consider the first-time demonstration of ferroelectric switching of BFO by the use of two different techniques a successful experimental challenge as well as a relevant advance in the knowledge of barium monoferrite.

Regarding the lack the spontaneous ferromagnetic moment, we again show from a symmetry perspective that the system possesses an instability toward a spiral/cycloidal state thanks to a Lifshitz invariant in the free energy. In the γ -BaFe₂O₄ case this term is overruled by competing interactions, but composition changes my tip the balance and induce more complex magnetic states. Moreover, as we highlight in the introduction, there are many applications in which AFM multiferroic materials are required.

For these reasons, we think that a low ferroelectric polarization and the absence of ferromagnetic canting do not make our material less interesting, especially from the point of view of fundamental physics.

Finally, concerning the possible presence of magnetoelectric coupling in the system, despite we agree that such observation would make BaFe₂O₄ even more interesting, we believe the present amount of data is suitable for publication, in particular as in our opinion the coexistence of magnetism and ferroelectricity at room temperature in a single phase compound is a relevant discover. Our present and future efforts are of course devoted to the search for proofs of magnetoelectric coupling.

REVIEWERS' COMMENTS

Reviewer #1 (Remarks to the Author):

I am satisfied with the answers of the authors and the corrections made to the article. I believe that the authors have done a serious job on the article and it can be considered for publication in the Nature Communications.

Reviewer #2 (Remarks to the Author):

The authors have completely addressed my concerns. I recommend for acceptance in the current form.

Reviewer #3 (Remarks to the Author):

The scientific motivation is clear in the current abstract. Now I support the publication of this manuscript in Nature Communications.